# Application of PEG-Covered Non-Biodegradable Polyelectrolyte Microcapsules in the Crustacean Circulatory System on the Example of the Amphipod *Eulimnogammarus verrucosus*

**DOI:** 10.3390/polym11081246

**Published:** 2019-07-27

**Authors:** Ekaterina Shchapova, Anna Nazarova, Anton Gurkov, Ekaterina Borvinskaya, Yaroslav Rzhechitskiy, Ivan Dmitriev, Igor Meglinski, Maxim Timofeyev

**Affiliations:** 1Institute of Biology, Irkutsk State University, 664025 Irkutsk, Russia; 2Baikal Research Centre, 664003 Irkutsk, Russia; 3Institute of Biology, Karelian Research Center of the Russian Academy of Sciences, 185910 Petrozavodsk, Russia; 4Optoelectronics and Measurement Techniques Laboratory, University of Oulu, 90570 Oulu, Finland; 5Aston Institute of Materials Research, School of Engineering & Applied Science, Aston University, Birmingham B4 7ET, UK; 6School of Life & Health Sciences, Aston University, Birmingham B4 7ET, UK

**Keywords:** biocompatibility, immunity, implantable sensors, invertebrate, layer-by-layer, primary cell cultures

## Abstract

Layer-by-layer assembled microcapsules are promising carriers for the delivery of various pharmaceutical and sensing substances into specific organs of different animals, but their utility in vivo inside such an important group as crustaceans remains poorly explored. In the current study, we analyzed several significant aspects of the application of fluorescent microcapsules covered by polyethylene glycol (PEG) inside the crustacean circulatory system, using the example of the amphipod *Eulimnogammarus verrucosus*. In particular, we explored the distribution dynamics of visible microcapsules after injection into the main hemolymph vessel; analyzed the most significant features of *E. verrucosus* autofluorescence; monitored amphipod mortality and biochemical markers of stress response after microcapsule injection, as well as the healing of the injection wound; and finally, we studied the immune response to the microcapsules. The visibility of microcapsules decreased with time, however, the central hemolymph vessel was confirmed to be the most promising organ for detecting the spectral signal of implanted microencapsulated fluorescent probes. One million injected microcapsules (sufficient for detecting stable fluorescence during the first hours after injection) showed no toxicity for six weeks, but in vitro amphipod immune cells recognize the PEG-coated microcapsules as foreign bodies and try to isolate them by 12 h after contact.

## 1. Introduction

Various implantable nano- and micro-scaled structures have great potential for becoming the foundation of novel technologies for monitoring and manipulating the state of diverse organisms. Such structures are, for example, polyelectrolyte microcapsules produced by the layer-by-layer (LbL) adsorption technique [1,2]. This technique is based on the deposition of some polymers onto various colloidal microparticles [3,4,5], including porous structures, such as calcium carbonate in the form of vaterite that can be preloaded with certain substances and further dissolved to obtain soft microcapsules with the substances inside [6,7]. These microcapsules can be easily prepared, using either biodegradable or non-biodegradable polymers to control their stability inside an organism for different applications [3]. Strongly charged poly(allylamine hydrochloride) (PAH) and poly(sodium 4-styrenesulfonate) (PSS) are among the most popular non-biodegradable polyelectrolytes used as sequential building blocks for microcapsule shells [8,9,10]. These two polymers form bilayers with a structure close to lamellar and can be used to prepare shells of given thickness [9].

For increasing the overall biocompatibility of the microcapsules, a polyethylene glycol (PEG) coating was suggested [11]. PEG can be attached to polyelectrolyte microcapsules in the form of a graft copolymer, with the charged main polymer sticking to the microcapsule surface electrostatically, while backbones of PEG obduce the microcapsule and reduce their aggregation and friction, which simplifies their distribution in the circulatory system [12,13], as well as decreasing protein adsorption to the surface, which can lower or at least postpone the immune response [11,13].

Crustaceans are an important source of aquatic food protein and popular research objects in ecophysiology [14,15]. However, possibilities for the application of microcapsules as an implantable tool for monitoring and adjusting the physiological status of crustaceans in vivo remain poorly explored. Previously, we applied the PEG-covered polyelectrolyte microcapsules to deliver the pH-sensitive fluorescent dye SNARF-1 into the main hemolymph vessel of the amphipod *Eulimnogammarus verrucosus* (Figure 1) and monitor pH changes in vivo [16]. The same approach is perspective for measuring other parameters, such as the concentration of different ions and metabolites [17], directly in the circulatory system of various crustaceans with translucent exoskeletons. The microcapsules used were composed of non-biodegradable PAH and PSS, to increase their stability and prolong the possibility of sensing.

In the present study, we analyze various nuances arising from the application of microencapsulated fluorescent dyes in the crustacean circulatory system, using the example of the same amphipod *E. verrucosus* (Figure 1), endemic to Lake Baikal [18]. In particular, we monitor the visibility of the PEG-coated PAH/PSS microcapsules after injection into the main hemolymph vessel and analyze the autofluorescence of *E. verrucosus* to highlight the most promising body parts for the application of microencapsulated molecular probes. Then, we assess the survival and stress response of the amphipods after injection of the microcapsules. Finally, we study the wound healing after injection and the amphipod immune response to the microcapsules.

## 2. Materials and Methods

### 2.1. Materials

All the chemicals used for the preparation of polyelectrolyte microcapsules and further procedures were of analytical grade and were applied without additional purification. The conjugate of fluorescein isothiocyanate with bovine serum albumin (FITC-albumin; #A9771) was purchased from Sigma-Aldrich (St. Louis, MO, USA), the conjugate of seminaphtharhodafluor-1 with dextran (SNARF-1-dextran; #D-3304) was bought from Thermo Fisher Scientific (Eugene, OR, USA). Polymers poly(allylamine hydrochloride) (PAH; #283215) and poly(sodium 4-styrenesulfonate) (PSS; #243051) were provided by Sigma-Aldrich (produced in USA and Belgium, respectively). The poly(L-lysine)-graft-poly(ethylene glycol) copolymer (PLL-g-PEG; #SZ34-67) was purchased from SuSoS (Dübendorf, Switzerland).

### 2.2. Preparation of Microcapsules

Fluorescent dyes were encapsulated at room temperature using the layer-by-layer adsorption of oppositely charged polyelectrolytes, as described previously [19]. The conjugates FITC-albumin and SNARF-1-dextran were co-precipitated in porous CaCO_3_ microcores by mixing 2 mL of 4 mg/mL conjugate solution with 0.615 mL of 1 M CaCl_2_ and 1 M Na_2_CO_3_ solutions. The cores were then covered with 12 layers of oppositely charged polyelectrolytes PAH and PSS and with the final layer of PLL-g-PEG. After dissolving CaCO_3_ cores in 0.1 M ethylenediaminetetraacetic acid solution (pH 7.1), the microcapsules had the following structure: dye conjugate/(PAH/PSS)_6_/PLL-g-PEG.

### 2.3. Animal Sampling and Maintenance

Adult individuals (~25–35 mm-long) of *Eulimnogammarus verrucosus* (Gerstfeldt, 1858) were caught with a hand net (kick sampling) in the Baikal littoral zone near the village of Listvyanka (51°52′13.3″ N 104°49′41.9″ E), at depths of 0–1.2 m. The amphipod species is not endangered or protected; no specific permission was required for sampling. During acclimation, amphipods were kept in well-aerated 3 L aquaria at a water temperature of approximately 6 °C, at a density of 15–20 individuals per aquarium. Amphipods were given one week to acclimatize to laboratory conditions before any experiments. The high levels of activity, feeding, and absence of mortality during the acclimation period confirmed that laboratory maintenance was not stressful for the amphipods.

All experimental procedures with amphipods were conducted in accordance with the EU Directive 2010/63/EU for animal experiments and were approved by the Animal Subjects Research Committee of Institute of Biology at Irkutsk State University.

### 2.4. Injection into the Circulatory System of Amphipods

The number of microcapsules being injected was one million per animal. The 2 μL suspension of microcapsules in the isotonic solution (0.9% aqueous solution of NaCl) was injected into the central hemolymph vessel of amphipods between the sixth and seventh segments of pereon using the IM-9B microinjector (Narishige, Tokyo, Japan). The injected microcapsules contained FITC-albumin, unless otherwise specified. During the injection, the individual was immobilized inside a wet polyurethane sponge with a temperature of approximately 6 °C.

### 2.5. Fluorescent Microscopy and Spectroscopy of Amphipods

Microcapsules and autofluorescence of *E. verrucosus* were visualized under the Mikmed-2 fluorescence microscope (LOMO, Saint Petersburg, Russia), with either the EOS 1200 camera (Canon, Taiwan) or the QE Pro spectrometer (Ocean Optics, Largo, FL, USA; acquisition range 350–1100 nm, INTSMA-200 optical slit, integration time 1–5 s) attached. The spectrometer was connected to the microscope, as described previously [19]. The animals were restrained under the 10× microscope objective using a thermostatic cell containing circulating Baikal water with a temperature of 5–8 °C, according to previous recommendations [20].

Fluorescent microscopy was performed in either the green channel (the peak excitation wavelength at 496 nm; long-pass emission filter from ~510 nm) or the red channel (the peak excitation wavelength at 546 nm; long-pass emission filter from ~585 nm). The parameters of the green channel were standard for visualization of FITC and similar dyes; the illumination conditions of the red channel were optimized to excite SNARF-1 in the used optical system [16,21]. Data analysis was performed using the Scilab package (www.scilab.org).

### 2.6. Determination of Biochemical Stress Response Markers

The biochemical measurements were performed in whole individuals frozen in liquid nitrogen. For the quantitative determination of 70 kDa heat shock proteins (HSP70), we used the method of Bedulina et al. [22]. The HSP70 content was determined by western blot. SDS electrophoresis was performed in polyacrylamide gel blocks using the Mini-PROTEAN II electrophoretic cell (Bio-Rad, USA). Relative molecular weights of proteins were estimated using the low molecular weight marker kit (Biomol, UK). Proteins were transferred to a polyvinylidene fluoride membrane, as described in Bedulina et al. [23]. For HSP70 assessment, the blots were incubated with primary antibody (anti-HSP70 antibody produced in mice; Sigma-Aldrich, #H5147) and then with secondary antibody (anti-mouse IgG:AP Conj.; Sigma-Aldrich #A3562). Bovine HSP70 (Sigma, #H9776) was used as a positive control. HSP70 levels were measured by the semiquantitative analysis of grey values on scanned western blot membranes, using the ImageJ software within the Fiji package [24].

Lactate levels were determined using the Lactate-Vital express kit (Vital–Diagnostics, St. Petersburg, Russia) according to Axenov-Gribanov et al. [25] Spectrophotometry was performed using the Cary 50 spectrophotometer (Varian, Palo Alto, CA, USA) at 505 nm (absorbance, here and further).

The activity of the enzyme glutathione S-transferase (GST; EC 2.5.10.13) was measured after extraction, according to Shchapova et al. [26]. The ratio of tissue homogenization buffer to amphipod biomass was 3:1 (v:w). Specimens were homogenized in 0.1 M sodium phosphate buffer (pH 6.5) and centrifuged at 10,000× *g* for 3 min. Glutathione S-transferase activity was measured using 0.97 mM 1-chloro-2,4-dinitrobenzene as substrate at 340 nm (pH 6.5) according to Habig et al. [27]. The Bradford assay was used to evaluate protein concentrations [28]. The assay was based on the binding of coomassie brilliant blue G-250 to residues of certain amino acids (mostly arginine and lysine), leading to an increase in absorbance at 595 nm [29]. Enzyme activities are expressed in nkat/mg protein.

### 2.7. Monitoring the Healing of the Injection Wound

The process of exoskeleton healing after the injection was monitored for four weeks. We defined three stages of repair based on the observed wound melanization [30]: no healing (no black coloration); healing is ongoing (indicated by the black coloration of the wound border); the exoskeleton is repaired. The external examination of the chitin state was followed by an internal checkup after the dissection of four–five individuals (euthanized with clove oil suspension) at each time point. Amphipods were examined under the SPM0880 stereomicroscope (Altami, Saint Petersburg, Russia).

### 2.8. Determination of the Phenoloxidase Activity

Phenoloxidase (PO) activity was measured in the amphipod hemolymph in response to microcapsule injection. Additionally, we tested the sensitivity of the PO system to yeast *Saccharomyces cerevisiae* as a positive control. Yeast cells resuspended in isotonic solution were injected as described for the microcapsules at concentrations of one million or six million cells per animal.

Hemolymph extracts were taken between the seventh and eighth dorsal segment using a sterile needle at 6 h, 12 h, 1 day, and 3 days after injection. Seventy microliters of hemolymph were collected into a sterile, pre-chilled glass capillary and placed into a 0.5 mL microcentrifuge tube containing 0.07 mL cold phosphate buffered saline (PBS; 150 mM NaCl, 10 mM Na_2_HPO_4_, pH 8) with 7 mg/mL phenylmethanesulfonyl fluoride, as modified from [31]. All samples were frozen in liquid nitrogen and stored at −80 °C until later measurements. After thawing, the samples were centrifuged for 10 min at 500 g and 4 °C to pellet the cell fraction.

About 10 µL hemolymph extract was mixed with 40 µL PBS, 280 µL distilled water, and 40 µL 3,4-dihydroxy-L-phenylalanine (L-DOPA; 4 mg/mL of distilled water) to measure the PO activity. The determination of PO activity was based on the catalytic conversion of L-DOPA to dopachrome, with the solution staining in red-brown color [32]. Measurements were performed with the Cary 50 spectrophotometer at a wavelength of 490 nm (absorbance) for 40 min. Enzyme activity was assessed as the slope of the reaction curve during the linear reaction phase [31].

### 2.9. Primary Culture of Amphipod Hemocytes

We established the primary culture of hemocytes isolated from the hemolymph of amphipods *E. verrucosus*. Before the hemolymph extraction, the dorsal side of the pereon surface was sterilized with 70% ethanol. The central hemolymph vessel was punctured with a sterile needle, and hemolymph was collected with a sterile glass capillary. The amphipod hemolymph was mixed (1:1) with the isotonic anticoagulant solution (150 mM NaCl, 5 mM Na_2_HPO_4_, 30 mM sodium citrate, 10 mM EDTA, pH 8.0; filtered through a 0.45 μm syringe filter) on ice to avoid undesired degranulation of granulocytes. Cells were separated by low-speed centrifugation (23 g for 2 min at 4 °C); the humoral fraction was discarded, and hemocytes were washed with a sterile buffer solution (150 mM NaCl, 5 mM Na_2_HPO_4_, pH 8.0), based on the available data on the hemolymph composition [33,34]. The washing procedure was repeated twice. The viability of amphipod hemocytes was assessed using vital staining with 0.4% trypan blue in a hemocytometer under the Mikmed-2 microscope. The used hemocyte extraction procedure resulted in at least 90% hemocyte viability, a median cell loss of approximately one-third of hemocytes, no increase in degranulation (as compared to the granule concentration in hemolymph), and aggregation of no more than 20% of the cells (aggregates were defined as a cluster of at least three hemocytes).

Hemocytes were suspended in sterile Leibovitz’s L-15 medium with L-glutamine (#1840516, Life Technologies, Belfast, UK), containing 15% fetal bovine serum (#FB-1001, Biosera, South America). The average volume of *E. verrucosus* hemolymph was 50–100 μL, and the used medium volume was 50 μL. The hemocyte concentration in the media was about 4000 cells/μL, which is approximately equal to the upper natural concentration in amphipod hemolymph. Cells were kept in 0.5 mL microtubes at 6–8 °C, with regular tube rotation to imitate the hemolymph circulation in the circulatory system of the animal. The viability of hemocytes in the primary culture did not drop below 90% for the three days.

### 2.10. Analysis of Hemocyte Aggregation after Contact with Microcapsules

The obtained primary culture of amphipod hemocytes was incubated with polyelectrolyte microcapsules or a suspension of *S. cerevisiae* cells, used as a positive control, to test the sensitivity of the culture to foreign bodies. Before the test, microcapsules or yeast cells were washed in sterile buffer (0.23 g for 5 min) and resuspended in the L-15 medium. Fifty microliter aliquots of the primary hemocyte culture, with approximately 0.2 million cells, were mixed with 2 μL aliquots containing either 0.2 million yeast cells or one million microcapsules (identical to the number injected during the other experiments). The recognition of the foreign bodies by hemocytes was assessed by measuring the hemocyte aggregation, defined as the proportion of free hemocytes compared to the initial total number of hemocytes in the sample. The aggregation process was examined at 1, 5, 12, 17, and 24 h after the start of incubation.

The staining of the hemocyte aggregates was performed with a dichlorotris(1,10-phenanthroline)ruthenium (II) hydrate (RuPhen_3_; Sigma-Aldrich, USA, #343714) and visualized in the green fluorescent channel. RuPhen_3_ contains aromatic components and may bind to hydrophobic structures of the cell, such as membrane lipids. Compounds with similar structure were used for the staining of nuclear components [35]. In the case of *E. verrucosus*, RuPhen_3_ was able to emphasize nuclei and granules of a significant part of hemocytes (but not all). Part of the microcapsules was also stained by RuPhen_3_ due to the interaction with PSS in the microcapsule shell [17].

### 2.11. Preparation of the Histological Section

For histological analysis, individual amphipods were sacrificed with deep anesthesia (in clove oil suspension) one week post injection of microcapsules containing SNARF-1-dextran. About 0.5 mL of Davidson’s solution was injected into the central vessel. Then, the urosome was removed, and crustaceans were completely immersed into Davidson’s solution for one day [36]. On the next day, the samples were rinsed, dissected to 0.5 mm-thick pieces and stored in 70% methyl alcohol. The specimens were embedded in paraffin, routinely using an STP 120 Spin Tissue Processor (Thermo Fisher Scientific, Walldorf, Germany), according to the protocol for tissues [37]. Tissues were cut into 6 µm-thick-slices using the HM-440 microtome (Thermo Fisher Scientific, Walldorf, Germany). The resulting sections were dewaxed and examined in the red fluorescent channel to detect microcapsules. Stained histological sections (hematoxylin and eosin) were studied in the bright field.

### 2.12. Statistical Analysis

The statistical significance of differences between the experimental and control (parallel if available or initial) groups was analyzed using the Mann–Whitney U test, with the Holm correction for multiple comparisons in R (www.r-project.org). The differences were considered significant with *p* < 0.05.

## 3. Results and Discussion

### 3.1. Visibility of the Microcapsules inside Amphipods: Distribution and Autofluorescence

The possibility of visualizing fluorescent microcapsules is of primary importance for such applications as physiological sensing in loco and contrasting various features of the crustacean circulatory system. Previously, we visualized the microcapsules only in the central hemolymph vessel of amphipods and did not analyze their behavior in other body parts [16]. The current study of the microcapsule visibility in the *E. verrucosus* body was carried out for six weeks, after the injection of one million microcapsules into the circulatory system. This number was chosen to obtain a stable, high concentration of microcapsules in the central hemolymph vessel for most individuals right after the delivery and was used throughout the whole study. The microcapsules contained the bright dye fluorescein (in the form of FITC-albumin) and were visualized in the green fluorescent channel.

Figure 1 demonstrates all body parts of *E. verrucosus* where the microcapsules were observed. Since chitin and tissues of *E. verrucosus* have sufficient translucency and mostly low autofluorescence intensity in the green channel, at least some microcapsules could be visualized in the circulatory system in all segments of the amphipod body. Figure 2 provides quantitative information on the distribution of visible microcapsules. The color key indicates the median concentrations of microcapsules per the area of the respective parts of the circulatory system.

During the first hours after injection, the observed median concentrations of microcapsules were among the highest in the central hemolymph vessel inside the first three segments of the amphipod pereon, but they decreased drastically already one week after the injection, and the microcapsules were not visible in more than half of the individuals (Figure 2). Thus, acquiring the spectral signal from microencapsulated optical probes inside the central hemolymph vessel is possible for the majority of injected individuals only during the first hours or days after injection. It is an important limitation that should be considered in experimental design involving the probe-carrying microcapsules. A similar decrease was observed in the case of the first segment of upper antennae, another potentially perspective organ for physiological sensing. However, the concentration of microcapsules in these segments was substantially lower than in the central hemolymph vessel; moreover, as the antennae are close to the amphipod eye, direct illumination for fluorescence excitation leads to anxiety for the animal.

Throughout the six weeks of the experiment, the highest concentration of visible microcapsules and the lowest rate of its decrease were observed in the amphipod head, and the median concentration in this area reached zero only at the end of the experiment (Figure 2). This fact opens intriguing possibilities for long-term measurements of various hormones in close vicinity to the central nervous system of crustaceans using implanted microsensors, as well as targeted delivery of some substances in this region using resolving microcapsules. However, similar to antennae, physiological sensing in the head will require the application of either animal anesthesia or luminescent sensors, which do not require any exciting illumination. The last body part where high microcapsule concentrations were found for most individuals is the amphipod urosome. However, this part of *E. verrucosus* has significant armor that limits the application of fluorescent sensors (see below), but it may be of interest for other species. Additionally, the immobilization of amphipods and other crustaceans by the end of the body can be less convenient than by the central body area [20], which also makes this segment less favorable.

Variability in the concentration of microcapsules in other body parts was shown to be too unstable to use at least half of individuals with injected microcapsules for physiological sensing. Nevertheless, special attention should be given to amphipod coxal plates (Figure 1), despite the median number of microcapsules in the plates never exceeding zero, for example, several dozens of microcapsules were visible in the first coxal plate of one third of individuals right after the injection. The distinctive feature of this body part is proximity of the hemolymph flow to the body surface. Thus, microcapsules in coxal plates can be visualized with the highest magnification possible among all non-moving parts of the amphipod body. This feature may allow local measurements of hemolymph parameters using the spectral signal from even single fluorescent microcapsule, while at least many dozens of them are required in the central hemolymph vessel [16].

The reasons for the gradual decrease in visibility of the PEG-covered microcapsules in the amphipod body remain unexplored. Despite the microcapsules being assembled of non-biodegradable polymers, they potentially could be decomposed to the individual polymers, for example, under extreme pH [38,39], which may be produced by the amphipod hemocytes [40] participating in the immune response. However, previous studies showed that cells of vertebrates seem to be unable to disintegrate the phagocytosed PAH/PSS-based microcapsules [21,41]. Additionally, the dissection of *E. verrucosus* after the six-week experiment showed plenty of intact fluorescent microcapsules around internal organs. So, they could simply migrate to deeper tissues and become invisible. The possibility of excretion of the microcapsules from the amphipod organism was not studied. It should also be noted that the microcapsules could become invisible due to the deposition of melanin pigments around them during the immune response [42]. In this case, they could stay at the same place in the circulatory system during the whole experiment, but become optically isolated.

We also studied the autofluorescence of the amphipods *E. verrucosus*, which may prevent the effective visualization of various fluorescent microsensors. The autofluorescence was visualized in the green and red channels, since excitation by short-wavelength light may be phototoxic [43] and is thus less useful for physiological studies. As mentioned already, the smooth parts of the amphipod body mostly have a relatively low background autofluorescence. Additionally, the autofluorescence spectra, especially in the green channel, have relatively low variability between individuals. Examples of such spectra for the first two pereon segments containing the central vessel of the circulatory system are depicted in Figure 3. The stability of the autofluorescence spectra (at least in the range 600–650 nm for the red channel) makes it possible to perform their subtraction from the spectral signal of the fluorescent microsensors, as has been shown previously [16].

However, there are two factors that significantly increase the intensity of autofluorescence in some parts of the amphipod body. First of all, sclerotized armor, such as spines and setae, has similar spectra, but enhanced intensity of autofluorescence in both channels. This is the case especially for the highly armored urosome of *E. verrucosus* (Figure 1), which makes separation of autofluorescence from the spectrum of microsensors difficult. Additionally, in the red channel, we observed pronounced autofluorescence in the area of the digestive system (Figure 1), which is approximately ten-fold higher than for the area of the central hemolymph vessel of the same pereon segment (Figure 3). The intense autofluorescence of the intestine in the red channel can contaminate the spectra of microsensors located both in the central hemolymph vessel and the coxal plates, and requires careful positioning of the amphipod under the objective.

The performed analyses allow us to conclude that despite the visibility of the PEG-coated polyelectrolyte microcapsules in the central hemolymph vessel of amphipods significantly decreasing within several days after injection, this organ is indeed the most promising for physiological monitoring using different implanted fluorescent microsensors inside the hemolymph of crustaceans. This is due to the combination of a number of factors: the convenience of injection into the vessel, easiness of fixation of the animal by this body part, high microcapsule concentration, sufficient remoteness from the eye and low level of autofluorescence.

### 3.2. Reaction of the E. verrucosus Organism to the Injection of Microcapsules

Another important aspect of the application of any substance being injected into the organism is toxicity. This is especially significant for non-biodegradable implants, which is most probably the case for the PAH/PSS-based microcapsules. The injection of microcapsules in the isotonic solution or saline alone revealed no effect on *E. verrucosus* survival in comparison to the parallel control group, without any injections, during six weeks (Figure 4). Similarly, no distinct effect of the same microcapsules was previously observed on the survival of the fish *Danio rerio* [21].

Additionally, we evaluated three biochemical markers of stress response after the injections (Figure 4): concentrations of lactate and heat shock proteins of the family HSP70, as well as activity of glutathione S-transferase (GST). GST activity is a widely used biomarker of intoxication, as it responds to various xenobiotics in many aquatic animals [44,45]. It was measured to assess the possible influence of microcapsule components after their potential disintegration (see above). Lactate is a product of anaerobic glycolysis activated under deficient oxygen supply [25], and its concentration was determined to disclose the possible local occlusions of the hemolymph flow caused by the microcapsules. HSP70 are considered by many authors as a sensitive biomarker that responds to a variety of stressors, including toxic effects and hypoxia [46,47,48]. GST activity and lactate concentrations monitored for two weeks, as well as HSP70 concentration evaluated for one week, showed no reaction (all *p* > 0.15) to the injections (Figure 4).

So, according to the results of our study, the chosen concentration of PEG-covered polyelectrolyte microcapsules demonstrates no lethal or significant sublethal effects on the amphipods *E. verrucosus*. The diameter of polyelectrolyte microcapsules was less than the size of crustacean hemocytes, and the appearance of currently unobserved toxic effects due to occlusions of the hemolymph flow is unexpected also for smaller crustaceans, unless a higher number of the microcapsules per hemolymph volume is used.

Since the integrity of amphipod integuments is an important factor in toxicological studies, we also studied the process of exoskeleton repair after the injections between the sixth and seventh segments of *E. verrucosus* pereon (Appendix A). The wound healing begins by the 10th day after the injection, and complete restoration of the chitin integrity was observed in 28 days from the day of the procedure. Since the visibility of microcapsules inside amphipods decreases faster than the exoskeleton can be restored, any toxicological experiments involving the implantation of the microencapsulated optical probes into the amphipod circulatory system will require artificial repair of the injection wound using, for example, some tissue adhesives.

### 3.3. Amphipod Immune Response to the Microcapsules

Finally, an important factor for the use of implantable sensors is the potential stimulation of the immune response, but the information on the possibility of immune reaction to implants with the PEG coating is fairly absent for invertebrates. The immune response of crustaceans is globally divided into the humoral and cellular branches [49]. The main humoral immune reaction that can be dangerous for the contact of any sensor with the internal environment (and thus its functionality) is melanization of the foreign body involving the enzyme phenoloxidase. Hemocytes are the cells circulating in crustacean hemolymph; they participate in the immune response by phagocytosis and aggregation around different foreign bodies [50], and thus can also isolate the sensor from the hemolymph.

The cells of the yeast *Saccharomyces cerevisiae* (similar in size to the microcapsules) were used as a positive control to test the sensitivity of *E. verrucosus* PO to foreign objects. Indeed, the median PO activity increased after the injection of yeast cells into the main hemolymph vessel, but a statistically significant difference from the parallel control group was observed only after the introduction of six million cells (*p* = 0.04) and not one million (Figure 5a). In the case of microcapsules (one million per animal; see above), there were no statistically significant differences from the control group (all *p* > 0.3) for three days after injection (Figure 5b).

The sensitivity of *E. verrucosus* hemocytes to the microcapsules was investigated in the primary culture in vitro, as was previously suggested for the red palm weevil *Rhynchophorus ferrugineus* [51]. It should be mentioned that the current study provides the first description of an established primary culture of hemocytes for amphipods in general. Again, yeast cells were used as the positive control and caused a statistically significant elevation in the hemocyte aggregation already in five hours (*p* = 0.01; all further *p* < 0.001) after introduction into the culture media (Figure 6a). The median proportion of hemocyte aggregation in the presence of yeast cells remained nearly the same from 5 to 24 h of the experiment.

Contact of the amphipod hemocytes with the microcapsules also triggered their aggregation, but at a lower rate (Figure 6a). In particular, a statistically significant difference from the parallel control group was observed only at 12 h (*p* < 0.003 here and further), while the median level of hemocyte aggregation reached the median level caused by yeast even later, at 24 h, although the introduced concentration of microcapsules was five times higher than the concentration of yeast cells. The microcapsules were observed inside the aggregates (Figure 6b), and their concentration in the culture media clearly decreased when the hemocyte aggregates appeared (data not shown). In several cases, we found a black coloration of the hemocyte aggregates that probably indicated the reaction of melanization (Appendix A). In order to partially visualize the membrane structures of hemocytes inside the aggregates, we applied the hydrophobic dye RuPhen_3_ with bright orange fluorescence. This staining clearly demonstrated that a significant proportion of hemocyte nuclei were located outside the cells (Figure 6b), which indicates the death of many cells during the immune reaction to the microcapsules.

In order to also verify the recognition of the microcapsules by *E. verrucosus* hemocytes in vivo, we performed a histological analysis of the amphipod body part containing the central hemolymph vessel for three individuals one week post injection. Indeed, we were able to find aggregates of microcapsules tightly enclosed by some cells, probably hemocytes, in two out of three samples (Appendix A). This fact, together with the observed in vitro melanization of hemocyte aggregates, also supports the hypothesis of decreasing microcapsule visibility due to immune reaction, at least to some extent. Additionally, it is more mild evidence against the possibility of decomposition of the PAH/PSS/PLL-g-PEG microcapsules to individual polymers inside crustaceans.

So, our results unambiguously demonstrate the existence of a cellular immune reaction of amphipods to the polyelectrolyte microcapsules, despite the PEG coating. Due to this effect, such microcapsules and similar probe-carrying microstructures composed of non-biodegradable polymers may become a promising tool for studying the behavior of immune cells in vivo. However, the stability of microcapsules inside the hemocyte aggregates makes them less suitable for prolonged (more than several hours) measurements of various hemolymph parameters, since it is hardly possible to distinguish the optical signals from free microcapsules and microcapsules within the hemocyte aggregates. Additionally, the stable microcapsules may attract more hemocytes and more resources for immune response than the biodegradable ones. Thus, the application of biodegradable polyelectrolyte microcapsules might be more recommended for physiological measurements inside crustaceans in vivo.

## 4. Conclusions

This study provides the first complex analysis of application of LbL-assembled polyelectrolyte microcapsules in the crustacean organism on the example of *E. verrucosus*. It is also the first study assessing the invertebrate immune response to PEG-covered implants. According to our results, the central hemolymph vessel should be proposed as the most convenient organ for visualization of fluorescent microsensors in most small crustaceans with translucent integument. The tested number of microcapsules (that was sufficient for stable visualization in the central hemolymph vessel) did not show lethal or sublethal toxic effects. Although the cellular immune response to the PEG-coated microcapsules was delayed in comparison to microorganisms, it also was the case. Nevertheless, during the first hours the proportion of involved hemocytes was low and could not affect a significant number of microcapsules. This makes the application of such microcapsules for physiological sensing possible, at least for the first several hours after injection.

## Figures and Tables

**Figure 1 polymers-11-01246-f001:**
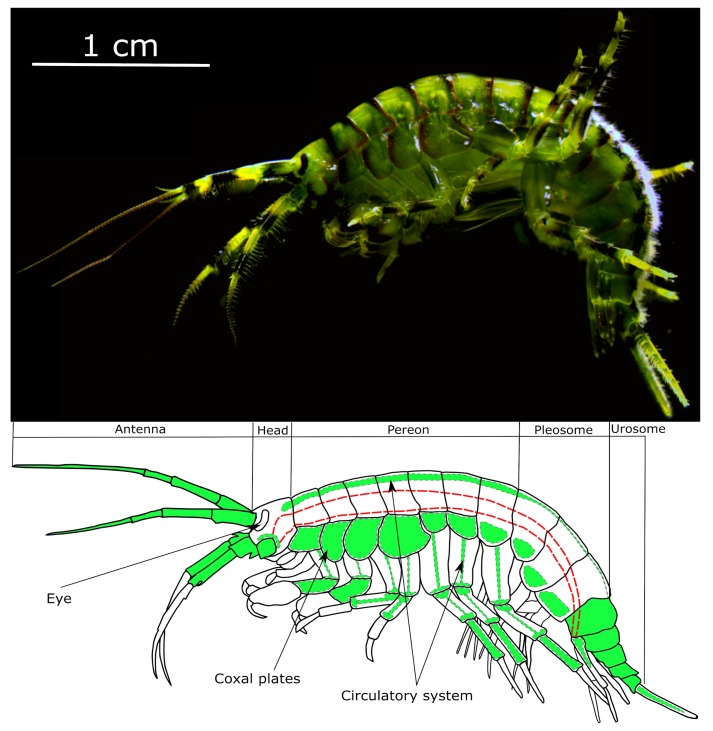
A photo of the amphipod *E**ulimnogammarus verrucosus* and schematic designation of all body parts where the microcapsules were observed (colored by green) after injection into the circulatory system. The red dashed line indicates the approximate position of the digestive system. The microcapsules contained fluorescein (in the form of FITC-albumin) and were monitored in the green channel.

**Figure 2 polymers-11-01246-f002:**
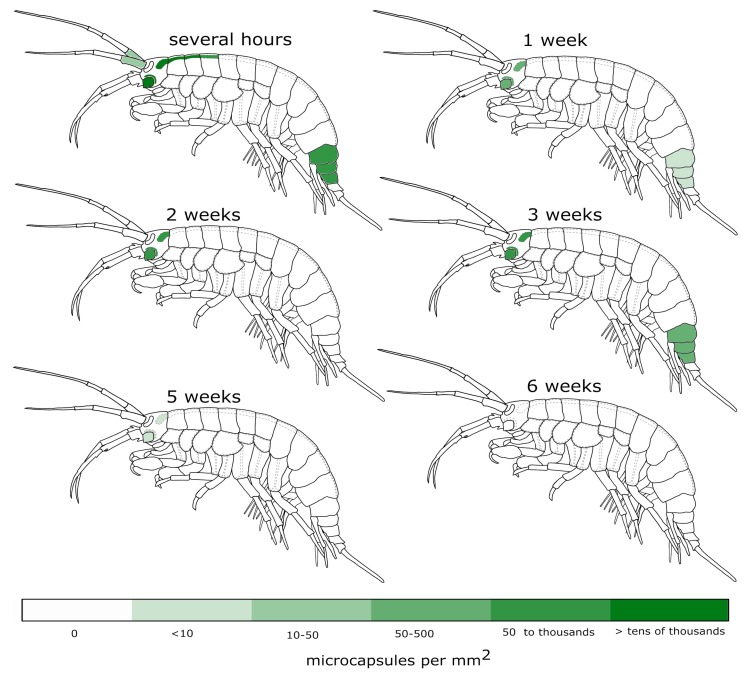
Monitoring of the visible microcapsules in different segments of *E. verrucosus* during the six weeks after injection. The microcapsules containing FITC-albumin were injected into the main hemolymph vessel (1 million per animal) and monitored in the green channel. The intensity of color indicates the median number of recognized microcapsules per area of the corresponding body part (*n* = 11–15 animals).

**Figure 3 polymers-11-01246-f003:**
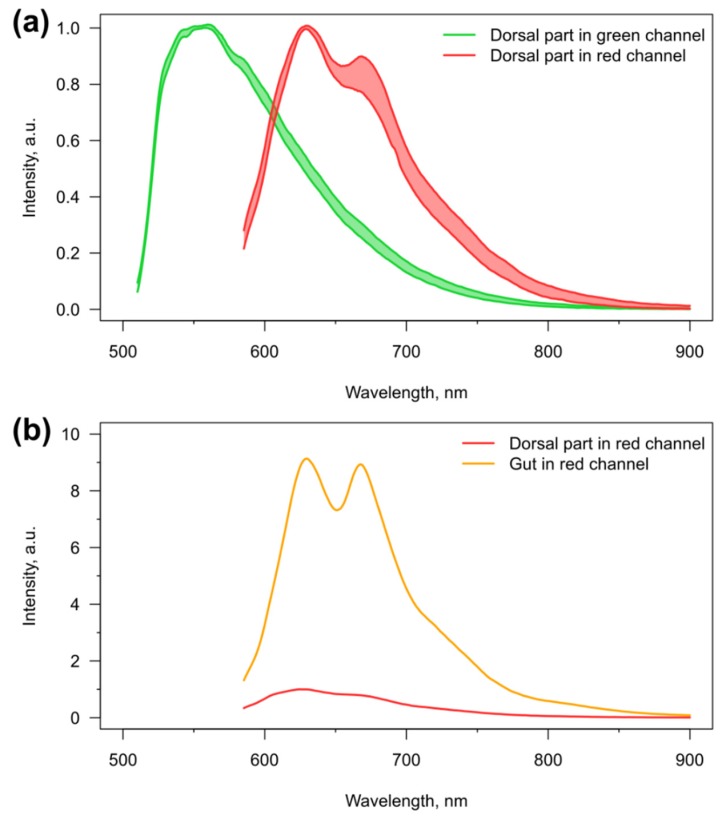
Autofluorescence of *E. verrucosus*. (**a**) Variability (*n* = 5) in autofluorescence spectra of the dorsal parts of the first two pereon segments in green and red channels, and (**b**) representative comparison of autofluorescence intensity between the dorsal and central (containing gut) parts of third pereon segment of the same individual in red channel. Spectra on the upper panel are aligned at the regions of peak intensity.

**Figure 4 polymers-11-01246-f004:**
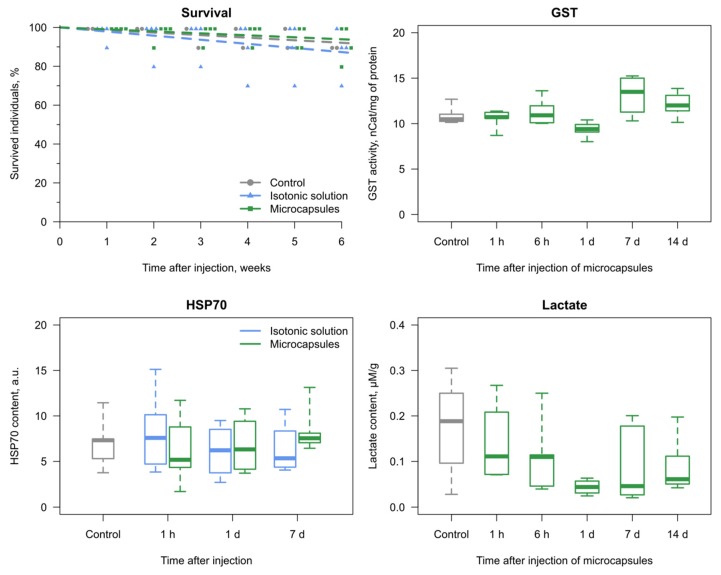
The reaction of the organism of amphipods *E. verrucosus* to the injection of 1 million microcapsules. The analyses included monitoring of mortality during six weeks (*n* = 10 per replicate; each point is a separate replicate); glutathione S-transferase activity (GST; *n* = 4–5) and lactate content (*n* = 5–7) for two weeks; and the content of heat shock proteins HSP70 (*n* = 4–6) for one week. No statistically significant differences from respective control groups (initial or parallel) were observed for the biochemical parameters.

**Figure 5 polymers-11-01246-f005:**
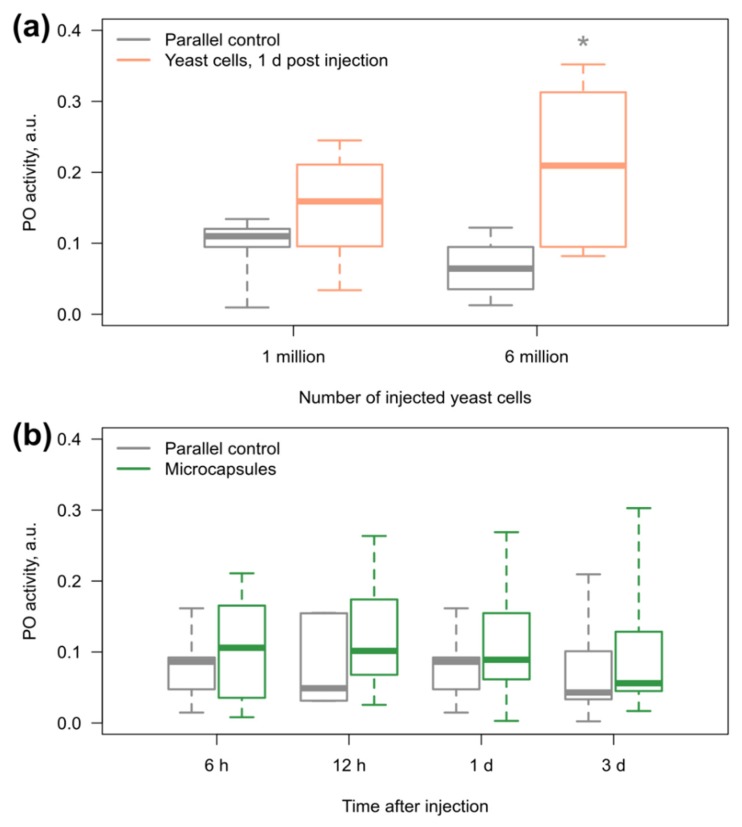
Hemolymph phenoloxidase (PO) activity of amphipods in response to injection of (**a**) yeast and (**b**) microcapsules. The injection of yeast cells (*n* = 6–8) was used as a positive control to verify the sensitivity of *E. verrucosus* PO to foreign bodies before monitoring of PO response to 1 million of the microcapsules (*n* = 5–11). * designates statistically significant difference from the parallel control group that did not receive any injections with *p* < 0.05.

**Figure 6 polymers-11-01246-f006:**
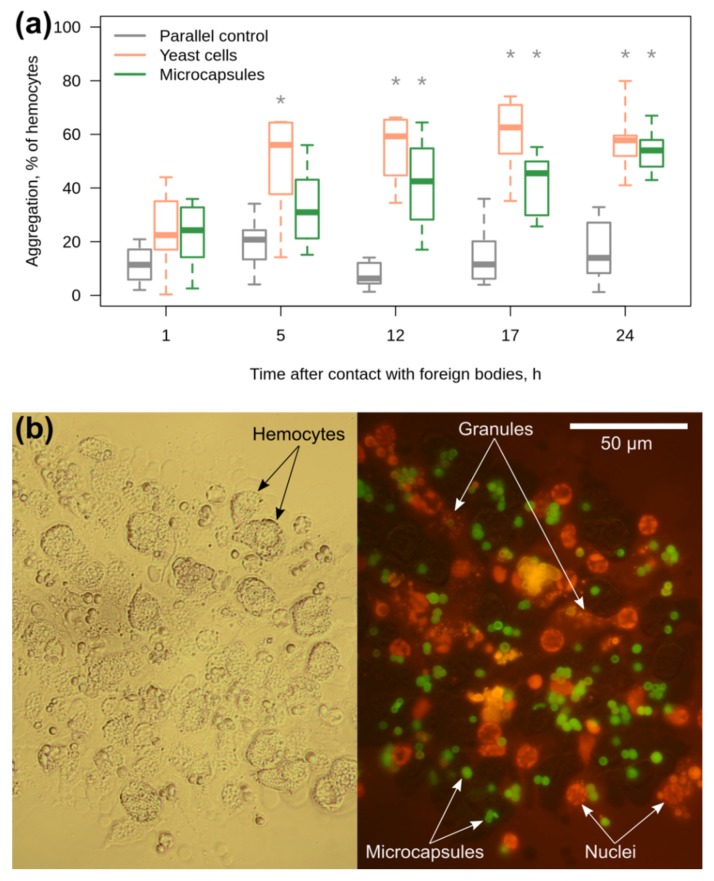
The reaction of primary culture of *E. verrucosus* hemocytes to yeast and microcapsules. (**a**) Monitoring of hemocyte aggregation after the introduction of yeast cells used as a positive control (1:1 of hemocytes) and microcapsules (5:1 of hemocytes) to the hemocyte media during 24 h (*n* = 8). (**b**) A representative example of a squashed aggregate of hemocytes with the microcapsules after the end of incubation in brightfield channel (left) and green fluorescent channel with the orange RuPhen_3_ staining (right). Microcapsules contained green FITC-albumin but partially obtained the yellow coloration due to contact with RuPhen_3_. * designates statistically significant difference from the parallel control group with *p* < 0.05.

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
