# Peer review of "Application of PEG-Covered Non-Biodegradable Polyelectrolyte Microcapsules in the Crustacean Circulatory System on the Example of the Amphipod Eulimnogammarus verrucosus"

_polymers, 2019, doi:10.3390/polym11081246_

Round 1
Reviewer 1 Report
the work by Shchapova et al. presents an interesting study on the application of LbL obtained capsules. The work is well driven and the results are sound. In my opinion, the work is suitable for Polymers. I have only a very minor comment:
-An interesting recent review on LbL multilayers as delivery platform was published by Guzmán et al. Authors should include it in the reference list.
Author Response
Dear Reviewer 1,
We would like to thank you for your comments.
> An interesting recent review on LbL multilayers as delivery platform was published by Guzmán et al. Authors should include it in the reference list.
The reference to the review by (Guzmán et al., 2017) is indeed important in the context of our manuscript. It has been included to the Introduction.
Very sincerely yours,
Dr. Sci., Prof. Maxim A. Timofeyev
Irkutsk State University
Reviewer 2 Report
Polyelectrolyte microcapsule is important carrier for drugs delivery. The manuscript entitled " Application of PEG-covered non-biodegradable 
 polyelectrolyte microcapsules in the crustacean 
 circulatory system on the example of the amphipod 
Eulimnogammarus verrucosus" contains novel data and is merit for readers. Nevertheless, some points should be addressed for further consideration.
Abstract should add more quantitative information. It is also avoid the "conclude".
Introduction part:
- Adsorption of polyelectrolyte layer by layer is many applications.
Papers are recommended to the authors: Polymers 10 (2), 220, 2018; Journal of Molecular Liquids, 110900,2019
- The introduction of PAH, PSS and PEG needs more details. Adsorption of PSS onto colloidal particles are throughly investigated:
Colloid and Polymer Science 297 (1), 13-22, 2019
- Fig. 1 and its capture must be in the same page.
3. Materials and methods
- Chemicals and the grade of chemicals should add in the Materials and Methods.
- Conditions for fluorescence need more information. Why the excitation wavelength of 496 and 546 nm are selected. How about emission wavelengths.
- Line 130, page 5, what kinds of spectrometry was measured. The absorbance or transmittance
- How to determine concentration of protein. Details of methods should be added.
4. Results and discussion
- A large space in page 6 after line 237 must be filled.
- The Intensity in Fig.3 maybe not correct. The unit isnot a.u because the experiment measurement is fluorescence not absorbance.
-The Alive individual against time after injection in Figure 4 is poor.
- Please check the unit of PO activity in Fig. 5.
- It is better to check the aggregation of hemocytes with the microcapsules by TEM.
5. Conclusions
Conclusions are too long but the originality isnot emphasized again.
6. References
References are not enough for this research
Author Response
Dear Reviewer 2,
We would like to thank you for the extensive examination of the manuscript and the comments provided.
Here are our point-by-point responses to the issues raised:
> Abstract should add more quantitative information. It is also avoid the "conclude".
The introduction part of the abstract was shortened to add more details describing results. We've added the most important information regarding the decrease in visibility of microcapsules with time and the suggestion of the central hemolymph vessel as the most convenient organ for signal detection from fluorescent microsensors. Quantitative details on the mortality experiment are also added.
> Introduction part:
> Adsorption of polyelectrolyte layer by layer is many applications. Papers are recommended to the authors: Polymers 10 (2), 220, 2018; Journal of Molecular Liquids, 110900,2019
The Introduction has been expanded, and these and some other important references were added.
> The introduction of PAH, PSS and PEG needs more details. Adsorption of PSS onto colloidal particles are throughly investigated: Colloid and Polymer Science 297 (1), 13-22, 2019
The details and additional references regarding the method and the chosen polymers are added.
> Fig. 1 and its capture must be in the same page.
Now it's corrected.
> 3. Materials and methods
> Chemicals and the grade of chemicals should add in the Materials and Methods.
The section describing the chemicals and their grade was added at the very beginning of Materials and Methods as suggested.
> Conditions for fluorescence need more information. Why the excitation wavelength of 496 and 546 nm are selected. How about emission wavelengths.
Thank you for pointing our attention to this issue; we've rebuilt the current section 2.5 to better explain the performed analysis of autofluorescence to the readers. Amphipod autofluorescence was acquired in the same conditions that are used for visualization of fluorescein (as well as probes based on this molecule) and SNARF-1 in the green and red channels respectively, and the excitation wavelengths 496 and 546 nm were chosen accordingly. Since the spectra of molecular probes are ought to be visualized using long-pass emission filters to obtain as much information as possible, autofluorescence of amphipods was analyzed the same way.
> Line 130, page 5, what kinds of spectrometry was measured. The absorbance or transmittance
All spectrophotometric procedures were used to measure the absorbance according to the cited techniques. The information is added in the current sections 2.6 and 2.8.
> How to determine concentration of protein. Details of methods should be added.
The Bradford technique is based on binding of coomassie brilliant blue to certain amino acid residues in the proteins and associated increase in absorbance approximately at 595 nm. The information is added to the text.
> 4. Results and discussion
> A large space in page 6 after line 237 must be filled.
Now it's corrected.
> The Intensity in Fig.3 maybe not correct. The unit isnot a.u because the experiment measurement is fluorescence not absorbance.
Indeed, the figure displays the fluorescence spectra. However, here our intention was to demonstrate comparative topology of the spectra on the upper panel and comparative intensity on the lower one. Thus, the spectra on the upper panel were aligned at the regions of peak intensity, while the intensities of spectra on the lower panel were divided by the peak intensity of the smaller spectrum ("Dorsal part...") to highlight the difference between two spectra. In the first case comparison of non-aligned spectra does not make any sense since ablsolute intensity of autofluorescence is highly varible depending on position of the animal, which constantly moves. In the second case use of original units provided by the spectrometer (so-called "counts") complicates understanding that difference between the displayed spectra is exactly 9-fold. Furthermore, original counts are also arbitrary, i.e. the performed modification of the units on the lower panel does not remove any information, but makes comparison of the spectra easier for readers.
> The Alive individual against time after injection in Figure 4 is poor.
Thank you for raising the issue; we changed the vertical axis label to 'Survived individuals, %'.
> Please check the unit of PO activity in Fig. 5.
The relevant studies on this topic utilized arbitrary units for PO, and we followed the same protocol: https://doi.org/10.1016/j.ijbiomac.2017.01.026, https://doi.org/10.1007/s00442-008-1211-y, https://doi.org/10.1051/apido/2010046. This technique is based on enzymatic conversion of colorless L-DOPA to red-brown dopachrome by PO. Unfortunately, our search for the coefficient of molar extinction of dopachrome showed that it seems to be unavailable, so we could not convert our measurements of PO activity to absolute katals and had to use arbitrary units instead.
> It is better to check the aggregation of hemocytes with the microcapsules by TEM.
Visualization of hemocyte aggregation using light microscopy is recommended in a number of papers: https://doi.org/10.1016/j.fsi.2012.11.035, https://doi.org/10.1111/1744-7917.12141,
https://doi.org/10.1016/j.toxicon.2014.02.006. Furthermore, use of the common TEM would not allow us to both quantify aggregation of cells and localize the fluorescent microcapsules between the cells within an aggregate at the same time. To our opinon, this is a significant drawback of using TEM.
> 5. Conclusions
> Conclusions are too long but the originality is not emphasized again.
The section of Conclusions was shorten and fully rebuilt to highlight the most important novelty of the study.
> 6. References
> References are not enough for this research
The reference list has been expanded.
Very sincerely yours,
Dr. Sci., Prof. Maxim A. Timofeyev
Irkutsk State University
Round 2
Reviewer 2 Report
The revised manuscript was significantly improved by the authors.
The paper can be accepted in Polymers with current form.